# Beyond Confidence Regions: Tight Bayesian Ambiguity Sets for Robust MDPs

**Reazul Hasan Russel**
Department of Computer Science
University of New Hampshire
rrussel@cs.unh.edu

**Marek Petrik**
Department of Computer Science
University of New Hampshire
mpetrik@cs.unh.edu

## Abstract

Robust MDPs (RMDPs) can be used to compute policies with provable worst-case guarantees in reinforcement learning. The quality and robustness of an RMDP solution are determined by the ambiguity set—the set of plausible transition probabilities—which is usually constructed as a multi-dimensional confidence region. Existing methods construct ambiguity sets as confidence regions using concentration inequalities which leads to overly conservative solutions. This paper proposes a new paradigm that can achieve better solutions with the same robustness guarantees without using confidence regions as ambiguity sets. To incorporate prior knowledge, our algorithms optimize the size and position of ambiguity sets using Bayesian inference. Our theoretical analysis shows the safety of the proposed method, and the empirical results demonstrate its practical promise.

## 1 Introduction

Markov decision processes (MDPs) provide a versatile framework for modeling reinforcement learning problems [4, 33, 38]. However, they assume that transition probabilities and rewards are known exactly which is rarely the case. Limited data sets, modeling errors, value function approximation, and noisy data are common reasons for errors in transition probabilities [16, 30, 45]. This results in policies that are brittle and fail when implemented. This is particularly true in the case of *batch* reinforcement learning [18, 20, 23, 32, 42].

A promising framework for computing *robust policies* is based on Robust MDPs (RMDPs). RMDPs relax the need for precisely known transition probabilities. Instead, transition probabilities can take on any value from a so-called *ambiguity set* which represents a set of plausible transition probabilities [9, 14, 24, 29, 32, 40, 46, 47]. RMDPs are also reminiscent of dynamic zero-sum games: the decision maker chooses the best actions, while the adversarial nature chooses the worst transition probabilities from the ambiguity set.

The practical utility of using RMDPs has been hindered by the lack of good ways of constructing ambiguity sets that lead to solutions that are robust without being too conservative. The standard approach to constructing ambiguity sets from concentration inequalities [1, 32, 42, 44] leads to theoretical guarantees but provides solutions that hopelessly conservative. Many problem-specific methods have been proposed too, but they are hard to use and typically lack finite-sample guarantees [3, 5, 16, 28].

The main contribution of this work is to introduce a new method for constructing ambiguity sets that are both significantly *less conservative* than existing ones [21, 32, 42] and also provide strong finite-sample guarantees. Similarly to some prior work on robust reinforcement learning and optimization, we use Bayesian assumptions to take advantage of domain knowledge which is often available [7, 8, 13, 47]. Our main innovation is to realize that the natural approach to building ambiguity sets as confidence intervals is unnecessarily conservative. Surprisingly, in the Bayesian setting, using a

95% confidence region for the transition probabilities is unnecessarily conservative to achieve 95% confidence in the robustness of the solution. We also derive new $L_1$ concentration inequalities of possible independent interest.

The remainder of the paper is organized as follows. Section 2 formally describes the framework and goals of the paper. Section 3 describes our main contribution, RSVF, a new method for constructing tight ambiguity sets from Bayesian models that are adapted to the optimal policy. We provide theoretical justification for the robustness of RSVF, but detailed theoretical analysis of its performance guarantees is beyond the scope of this work. Then, Section 4 overviews related work and outlines methods that build ambiguity sets as frequentist confidence regions or Bayesian credible sets. Finally, Section 5 presents empirical results on several problem domains.

## 2 Problem Statement: Data-driven RMDPs

This section formalizes our goals and reviews relevant results for robust Markov decision processes (RMDPs). Throughout the paper, we use the symbol $\Delta^S$ to denote the probability simplex in $\mathbb{R}_+^S$. The symbols $\mathbf{1}$ and $\mathbf{0}$ denote vectors of all ones and zeros, respectively, of an appropriate size. The symbol $\mathbf{I}$ represents the identity matrix.

### 2.1 Safe Return Estimate: VaR

The underlying reinforcement learning problem is a Markov decision process with states $\mathcal{S} = \{1, \ldots, S\}$ and actions $\mathcal{A} = \{1, \ldots, A\}$. The rewards $r : \mathcal{S} \times \mathcal{A} \to \mathbb{R}$ are known but the true transition probabilities $P^\star : \mathcal{S} \times \mathcal{A} \to \Delta^S$ are unknown. The transition probability vector for a state $s$ and an action $a$ is denoted by $p^\star_{s,a}$. As this is a *batch* reinforcement learning setting, a fixed dataset $\mathcal{D}$ of transition samples is provided: $\mathcal{D} = (s_i \in \mathcal{S}, a_i \in \mathcal{A}, s'_i \in \mathcal{S})_{i=1,\ldots,m}$. The only assumption about $\mathcal{D}$ is that the state $s'$ in $(s, a, s') \in \mathcal{S}$ is distributed according to the *true* transition probabilities $s' \sim P^\star(s, a, \cdot)$, no assumptions are made on the sampling policy. Note that in the Bayesian approach, $P^\star$ is a random variable and we assume to have a prior distribution available.

The objective is to maximize the standard $\gamma$-discounted infinite horizon return [33]. Because this paper analyzes the impact of using different transition probabilities, we use a subscript to indicate which ones are used. The optimal value function for some transition probabilities $P$ is, therefore, denoted as $v_P^\star : \mathcal{S} \to \mathbb{R}$, and the value function for a *deterministic policy* $\pi : \mathcal{S} \to \mathcal{A}$ is denoted as $v_P^\pi$. The set of all deterministic stationary policies is denoted by $\Pi$. The total return $\rho(\pi, P)$ of a policy $\pi$ under transition probabilities $P$ is:

$$\rho(\pi, P) = p_0^\mathsf{T} v_P^\pi,$$

where $p_0$ is the initial distribution.

Ideally, we could compute a policy $\pi : \mathcal{S} \to \mathcal{A}$ that maximizes the return $\rho(\pi, P^\star)$, but $P^\star$ is unknown. Ignoring the uncertainty in $P^\star$ completely leads to brittle policies. Instead, a common objective in robust reinforcement learning is to maximize a plausible *lower-bound* on the return. Having a safe return estimate is very important since it can inform the stakeholder that the policy may not be good enough when deployed. The objective of computing a policy $\pi$ that maximizes a *high-confidence* lower bound on the return can be expressed as [8, 21, 31, 42]:

$$\max_{\pi \in \Pi} \text{V@R}_{P^\star}^\delta [\rho(\pi, P^\star)], \tag{1}$$

where $\text{V@R}^\delta$ is the popular value-at-risk measure at a risk level $\delta$ [35]. This objective is also sometimes known as *percentile optimization* [8]. It is important to note that the risk metric is applied over possible values of the uncertain parameter and not over the distribution of returns. For example, if $\text{V@R}_{P^\star}^{0.05}[\rho(\pi, P^\star)] = -1$ then for 5% of uncertain transition probabilities $P^\star$, the return is $-1$ or smaller.

Because solving the optimization problem in (1) is NP-hard [8], we instead maximize a lower bound $\tilde{\rho}(\pi)$. We call this lower bound a *safe return estimate* and it is defined as follows.

**Definition 2.1** (Safe Return Estimate). The estimate $\tilde{\rho} : \Pi \to \mathbb{R}$ of return is called *safe* for a policy $\pi$ with probability $1 - \delta$ if $\tilde{\rho}(\pi) \leq \text{V@R}_{P^\star}^\delta[\rho(\pi, P^\star)]$, or in other words if it satisfies:

$$\mathbb{P}_{P^\star}\left[\tilde{\rho}(\pi) \leq \rho(\pi, P^\star) \,\middle|\, \mathcal{D}\right] \geq 1 - \delta .$$

Recall that under Bayesian assumptions, $P^\star$ is a random variable and the guarantees are conditional on the dataset $\mathcal{D}$. This is different from the frequentist approach, in which the random variable is $\mathcal{D}$ and the guarantees are conditional on $P^\star$. The relative merits of Bayesian versus frequentist approaches to robust optimization have been discussed in earlier work [8, 47], but we emphasize that each approach presents a different set of advantages. An insightful discussion of the differences between the two approaches can be found, for example, in Sections 5.2.2 and 6.1.1 of Murphy (2012).

The following example will be used throughout the paper to demonstrate the proposed methods and visualize simple ambiguity sets.

*Example* 2.1. Consider an MDP with 3 states: $s_1, s_2, s_3$ and a single action $a_1$. Assume that the true, but unknown, transition probability is $P^\star(s_1, a_1, \cdot) = [0.3, 0.2, 0.5]$. The known prior distribution over $p^\star_{s_1, a_1}$ is Dirichlet with concentration parameters $\alpha = (1, 1, 1)$. The dataset $\mathcal{D}$ is comprised of 3 occurrences of transitions $(s_1, a_1, s_1)$, 2 of transitions $(s_1, a_1, s_2)$, and 5 of transitions $(s_1, a_1, s_3)$. The posterior distribution over $p^\star_{s_a, a_1}$ is also Dirichlet with $\alpha = (4, 3, 6)$. Note that this is a probability distribution over transition probability distributions. Fig. 1 depicts the posterior distribution projected onto the probability simplex along with a 90% confidence region centered on the posterior mean.

## 2.2 Robust MDPs

Robust Markov Decision Processes (RMDPs) are a convenient model and tractable model that generalizes MDPs. We will use RMDPs to maximize a tractable lower bound on V@R objective in (1) and compute a *safe* return estimate. Our RMDP model has the same states $\mathcal{S}$, actions $\mathcal{A}$, rewards $r_{s,a}$ as the MDP. The transition probabilities for each state $s$ and action $a$, denoted as $p_{s,a} \in \Delta^S$, are assumed chosen adversarialy from an *ambiguity set* $\mathcal{P}_{s,a}$. We use $\mathcal{P}$ to refer cumulatively to $\mathcal{P}_{s,a}$ for all states $s$ and actions $a$.

We restrict our attention to sa-rectangular ambiguity sets, which allow the adversarial nature to choose the worst transition probability independently for each state and action [22, 45]. Limitations of rectangular ambiguity sets are known well [12, 25, 43] but they represent a simple, tractable, and practical model. A convenient way of defining ambiguity sets is to use a norm-distance from a given *nominal transition probability* $\bar{p}_{s,a}$:

$$\mathcal{P}_{s,a} = \left\{ p \in \Delta^S \ : \ \|p - \bar{p}_{s,a}\|_1 \leq \psi_{s,a} \right\} \tag{2}$$

for a given $\psi_{s,a} \geq 0$ and a nominal point $\bar{p}_{s,a}$. We focus on ambiguity sets defined by the $L_1$ norm because they give rise to RMDPs that can be solved very efficiently [15].

RMDPs have properties that are similar to regular MDPs (see, for example, [2, 19, 22, 28, 45]). The robust Bellman operator $\widehat{T}_\mathcal{P}$ for an ambiguity set $\mathcal{P}$ for a state $s$ computes the best action with respect to the worst-case realization of the transition probabilities:

$$(\widehat{T}_\mathcal{P} v)(s) := \max_{a \in \mathcal{A}} \ \min_{p \in \mathcal{P}_{s,a}} (r_{s,a} + \gamma \cdot p^\mathsf{T} v) \tag{3}$$

The symbol $\widehat{T}_\mathcal{P}^\pi$ denotes a robust Bellman update for a given *stationary* policy $\pi$. The optimal robust value function $\hat{v}^\star$, and the robust value function $\hat{v}^\pi$ for a policy $\pi$ are unique and must, similarly to MDPs, satisfy $\hat{v}^\star = \widehat{T}_\mathcal{P} \hat{v}^\star$ and $\hat{v}^\pi = \widehat{T}_\mathcal{P}^\pi \hat{v}^\pi$. In general, we use a hat to denote quantities in the RMDP and omit it for the MDP. When the ambiguity set $\mathcal{P}$ is not obvious from the context, we use it as a subscript $\hat{v}^\star_\mathcal{P}$. The robust return $\hat{\rho}$ is defined as [16]:

$$\hat{\rho}(\pi, \mathcal{P}) = \min_{P \in \mathcal{P}} \rho(\pi, P) = p_0^\mathsf{T} \hat{v}^\pi_\mathcal{P} \ ,$$

where $p_0 \in \Delta^S$ is the initial distribution. In the remainder of the paper, we describe methods that construct $\mathcal{P}$ from $\mathcal{D}$ in order to guarantee that $\hat{\rho}$ is a tight lower bound on V@R of the returns.

## 3 Optimized Bayesian Ambiguity Sets

In this section, we describe the new algorithm for constructing Bayesian ambiguity sets that can compute less-conservative lower bounds on the return. RSVF (robustification with sensible value functions) is a Bayesian method that uses samples from the posterior distribution over $P^\star$ to construct tight ambiguity sets.

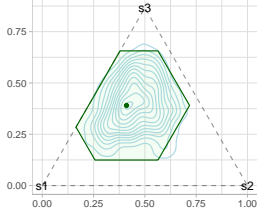
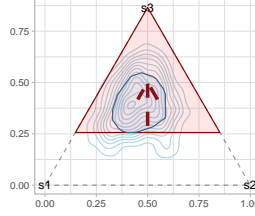
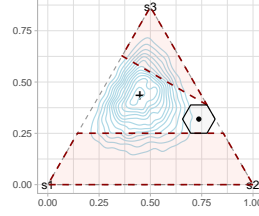

Figure 1: Contours of the posterior distribution and the 90%-confidence region.

Figure 2: Optimal Bayesian ambiguity set (red) for a value function $v = (0, 0, 1)$.

Figure 3: Sets $\mathcal{K}_{s_1,a_1}(v_i)$ (dashed red) for $i = 1, 2$ and $\mathcal{L}_{s_1,a_1}(\{v_1, v_2\})$ (black).

Before describing the algorithm, we use the setting of Example 2.1 to motivate our approach. To minimize distractions by technicalities, assume that the goal is to compute the return for a *single* time step starting from state $s_1$. Assume also that the value function $v = (1, 0, 0)$ is known, all rewards from $s_1$ are 0, and $\gamma = 1$. Recall that our goal is to construct a safe return estimate $\tilde{\rho}(\pi)$ of $\mathrm{V@R}^{0.1}_{P^\star}[\rho(\pi, P^\star)]$ at the 90% level. When the value function is known, it is possible to construct the *optimal* ambiguity set $\mathcal{P}^\star$ such that $\hat{\rho}(\pi) = \min_{p \in \mathcal{P}^\star} p^\mathsf{T} v = \mathrm{V@R}^{0.1}_{P^\star}[\rho(\pi, P^\star)]$ as:

$$\mathcal{P}^\star = \left\{ p \in \Delta^3 \; : \; p^\mathsf{T} v \geq \mathrm{V@R}^{0.1}_{P^\star}[\rho(\pi, P^\star)] \right\} .$$

It can be shown readily that this ambiguity set is optimal in the sense that any set for which $\tilde{\rho}(\pi)$ is exact must be a subset of $\mathcal{P}^\star$ [13]. Fig. 2 depicts the optimal ambiguity set along with the arrow that indicates the direction along which $v$ increases.

The optimal ambiguity set described above cannot be used directly, unfortunately, because the value function is unknown. It would be tempting to construct the ambiguity set as the *intersection* of optimal sets for all possible value functions; a polyhedral approximation of this set is shown in Fig. 2 using a blue color. Unfortunately, this approach is not (usually) correct and will not lead to a safe return estimate. This can be shown from the fact that support functions to convex sets are convex and V@R is not a convex (concave) function [6, 34]; see Gupta (2015) for a more detailed discussion.

Since it is not possible, in general, to simply consider the intersection of optimal ambiguity sets for all possible value functions, we approximate the optimal ambiguity set for a few reasonable value functions. For this purpose, we use a set $\mathcal{K}_{s,a}(v)$ defined as follows:

$$\mathcal{K}_{s,a}(v) = \left\{ p \in \Delta^S \; : \; p^\mathsf{T} v \leq \mathrm{V@R}^{\zeta}_{P^\star} \left[ (p^\star_{s,a})^\mathsf{T} v \right] \right\} ,$$

where $\zeta = 1 - \delta/(SA)$. The bottom dashed set in Fig. 3 depicts this set $\mathcal{K}$ for $v = (0, 0, 1)$ in Example 2.1. The intuition behind this construction is as follows. If any ambiguity set $\mathcal{P}_{s,a}$ intersects $\mathcal{K}_{s,a}(\hat{v}^\pi_{\mathcal{P}})$ for each state $s, a$ then the value function $\hat{v}^\pi_{\mathcal{P}}$ is safe: $\max_{p \in \mathcal{K}_{s,a}(v)} p^\mathsf{T} v \leq \mathrm{V@R}^{\zeta}_{P^\star} \left[ (p^\star_{s,a})^\mathsf{T} v \right]$. See Lemma B.2 for the formal statement.

The set $\mathcal{K}_{s,a}(v)$ is sufficient, when the value function is known, but we need to generalize the approach to unknown value functions. The set $\mathcal{L}_{s,a}(\mathcal{V})$ provides such a guarantee for a set of possible value functions (POV) $\mathcal{V}$. Its center is chosen to minimize its size while intersecting $\mathcal{K}_{s,a}(v)$ for each $v$ in $\mathcal{V}$ and is constructed as follows.

$$\begin{aligned}
\mathcal{L}_{s,a}(\mathcal{V}) &= \left\{ p \in \Delta^S \; : \; \|p - \theta_{s,a}(\mathcal{V})\|_1 \leq \psi_{s,a}(\mathcal{V}) \right\} \\
\psi_{s,a}(\mathcal{V}) &= \min_{p \in \Delta^S} f(p), \quad \theta_{s,a}(\mathcal{V}) \in \arg \min_{p \in \Delta^S} f(p), \quad f(p) = \max_{v \in \mathcal{V}} \min_{q \in \mathcal{K}_{s,a}(v)} \|q - p\|_1
\end{aligned} \tag{4}$$

The optimization in (4) can be represented and solved as a linear program. Fig. 3 shows the set $\mathcal{L}$ in black solid color. It is the smallest $L_1$-constrained set that intersects the two $\mathcal{K}$ sets for value functions $v_1 = (0, 0, 1)$ and $v_2 = (2, 1, 0)$ in Example 2.1.

We are now ready to describe RSVF, which is outlined in Algorithm 1. RSVF takes an optimistic approach to approximating the optimal ambiguity set. It starts with a small set of potential optimal value functions (POV) and constructs an ambiguity set that is safe for these value functions. It keeps increasing the POV set until $\hat{v}^\star$ is in the set and the policy is safe. To simplify presentation,

---
**Algorithm 1:** RSVF: Adapted Ambiguity Sets
---
    **Input:** Confidence $1 - \delta$ and posterior $\mathbb{P}_{P^\star}[\cdot \mid \mathcal{D}]$
    **Output:** Policy $\pi$ and lower bound $\tilde{\rho}(\pi)$
**1** $k \leftarrow 0$;
**2** Pick some initial value function $\hat{v}_0$;
**3** Initialize POV: $\mathcal{V}_0 \leftarrow \emptyset$ ;
**4** **repeat**
**5**     Augment POV: $\mathcal{V}_{k+1} \leftarrow \mathcal{V}_k \cup \{v_k\}$ ;
**6**     For all $s, a$ update $\mathcal{P}_{s,a}^{k+1} \leftarrow \mathcal{L}_{s,a}(\mathcal{V}_{k+1})$ ;
**7**     Solve $\hat{v}_{k+1} \leftarrow \hat{v}_{\mathcal{P}_{k+1}}^{\star}$ and $\hat{\pi}_{k+1} \leftarrow \hat{\pi}_{\mathcal{P}_{k+1}}^{\star}$;
**8**     $k \leftarrow k + 1$ ;
**9** **until** *safe for all* $s, a$: $\mathcal{K}_{s,a}(\hat{v}_k) \cap \mathcal{P}_{s,a}^k \neq \emptyset$;
**10** **return** $(\hat{\pi}_k, p_0^\mathsf{T} \hat{v}_k)$ ;
---

Algorithm 1 is not guaranteed to terminate in finite time; the actual implementation switches to BCI described in Section 4.2 after 100 iterations, which guarantees its termination.

The following theorem states that Algorithm 1 produces a safe estimate of the true return.

**Theorem 3.1.** *Suppose that Algorithm 1 terminates with a policy* $\hat{\pi}_k$ *and a value function* $\hat{v}_k$ *in the iteration* $k$. *Then, the return estimate* $\tilde{\rho}(\hat{\pi}) = p_0^\mathsf{T} \hat{v}_k$ *is safe:* $\mathbb{P}_{P^\star} \left[ p_0^\mathsf{T} \hat{v}_k \leq p_0^\mathsf{T} v_{P^\star}^{\hat{\pi}_k} \,\middle|\, \mathcal{D} \right] \geq 1 - \delta$.

Before discussing the proof of Theorem 3.1, it is important to mention its limitations. This result shows only that the return estimate $\hat{\rho}$ is safe; it does not show that it is good. There are, of course, naive safe estimates such as $\tilde{\rho}(\pi) = (1 - \gamma)^{-1} \min_{s,a} r_{s,a}$. Since RSVF tightly approximates the optimal ambiguity sets, we expect it to perform significantly better. The theoretical analysis of this of the approximation error of $\hat{\rho}$ is beyond the scope of this work and we present empirical evidence in Section 5 instead.

All proofs can be found in Appendix B. The proof is technical but conceptually simple. It is based on two main properties. The first one is the construction of optimal ambiguity sets for the known value function as outlined above. The second is the fact that the ambiguity set needs to be robust with only with respect to the robust value function $\hat{v}$ and *not* the optimal value function $v^\star$. This is subtle, but *crucial* since $\hat{v}$ is a constant while $v^\star$ is a random variable in the Bayesian setting. The RSVF approach, therefore, does not work when frequentist guarantees are required. Confidence regions, described in Section 4, are designed for situations when robustness is required with respect to a random variable, and are therefore overly conservative in our setting. See Appendix E for more in-depth discussion.

# 4 Ambiguity Sets as Confidence Regions

In this section, we describe the standard approach to constructing ambiguity sets as multidimensional confidence regions and propose its extension to the Bayesian setting. Confidence regions derived from concentration inequalities have been used previously to compute bounds on the true return in off-policy policy evaluation [41, 42]. These methods, unfortunately, do not readily generalize to the policy optimization setting, which we target. Other work has focused on reducing variance rather than on high-probability bounds [18, 23, 26]. Methods for exploration in reinforcement learning, such as MBIE or UCRL2, also construct ambiguity sets using concentration inequalities [10, 17, 37, 37, 39] and compute optimistic (upper) bounds to guide exploration.

## 4.1 Distribution-free (Frequentist) Confidence Interval

Distribution-free confidence regions are used widely in reinforcement learning to achieve robustness [32, 42] and to guide exploration [36, 39]. The confidence region is constructed around the mean transition probability by combining the Hoeffding inequality with the union bound [32, 44].

We refer to this set as a *Hoeffding confidence region* and define it as follows for each $s$ and $a$:

$$\mathcal{P}_{s,a}^H = \left\{ p \in \Delta^S : \|p - \bar{p}_{s,a}\|_1 \leq \sqrt{\frac{2}{n_{s,a}} \log \frac{SA2^S}{\delta}} \right\},$$

where $\bar{p}_{s,a}$ is the mean transition probability computed from $\mathcal{D}$ and $n_{s,a}$ is the number of transitions in $\mathcal{D}$ originating from state $s$ and an action $a$.

**Theorem 4.1.** *The robust value function $\hat{v}_{\mathcal{P}^H}$ for the ambiguity set $\mathcal{P}^H$ satisfies:*

$$\mathbb{P}_{\mathcal{D}}\left[\hat{v}_{\mathcal{P}^H}^\pi \leq v_{P^\star}^\pi, \ \forall \pi \in \Pi \mid P^\star\right] \geq 1 - \delta . \tag{5}$$

*In addition, if $\hat{\pi}_{\mathcal{P}^H}^\star$ is the optimal solution to the RMDP, then $p_0^\mathsf{T} \hat{v}_{\mathcal{P}^H}^\star$ is a* safe *return estimate of $\hat{\pi}_{\mathcal{P}^H}^\star$.*

To better understand the limitations of using concentration inequalities, we compare with new, and significantly tighter, frequentist ambiguity sets. The size of $\mathcal{P}^H$ grows as a square root of the number of states because of the $2^S$ term. This means that the size of $\mathcal{D}$ must scale about quadratically with the number of states to achieve the same confidence. Under some restrictive assumptions, the ambiguity set can be shown to be:

$$\mathcal{P}_{s,a}^M = \left\{ p \in \Delta^S \ : \ \|p - \bar{p}_{s,a}\|_1 \leq \sqrt{\frac{2}{n_{s,a}} \log \frac{S^2 A}{\delta}} \right\} .$$

This auxiliary result is proved in Appendix C.1. We emphasize that the aim of this bound is to understand the limitations of distribution free bounds, and we use it even when the necessary assumptions are violated.

## 4.2 Bayesian Credible Region (BCI)

We now describe how to construct ambiguity sets from Bayesian credible (or confidence) regions. To the best of our knowledge, this approach has not been studied explicitly. The construction starts with a (hierarchical) Bayesian model that can be used to sample from the posterior probability of $P^\star$ given data $\mathcal{D}$. The implementation of the Bayesian model is irrelevant as long as it generates posterior samples efficiently. For example, one may use a Dirichlet posterior, or use MCMC sampling libraries like JAGS, Stan, or others [11].

The posterior distribution is used to optimize for the *smallest* ambiguity set around the mean transition probability. Smaller sets, for a fixed nominal point, are likely to result in less conservative robust estimates. The BCI ambiguity set is defined as follows:

$$\mathcal{P}_{s,a}^B = \left\{ p \in \Delta^S \ : \ \|p - \bar{p}_{s,a}\|_1 \leq \psi_{s,a}^B \right\}, \quad \bar{p}_{s,a} = \mathbb{E}_{P^\star}[p_{s,a}^\star \mid \mathcal{D}] .$$

There is no closed-form expression for the Bayesian ambiguity set size. It must be computed by solving the following optimization problem for each state $s$ and action $a$:

$$\psi_{s,a}^B = \min_{\psi \in \mathbb{R}_+} \left\{ \psi \ : \ \mathbb{P}\left[\|p_{s,a}^\star - \bar{p}_{s,a}\|_1 > \psi \mid \mathcal{D}\right] < \frac{\delta}{SA} \right\} .$$

The nominal point $\bar{p}_{s,a}$ is fixed (not optimized) to preserve tractability. This optimization problem can be solved by the Sample Average Approximation (SAA) algorithm [35]. Algorithm 2, in the appendix, summarizes the sort-based method. The main idea is to sample from the posterior distribution and then choose the minimal size $\psi_{s,a}$ that satisfies the constraint. We assume that it is possible to draw enough samples from $P^\star$ that the sampling error becomes negligible. Because the finite-sample analysis of SAA is simple but tedious, we omit it.

**Theorem 4.2.** *The robust value function $\hat{v}_{\mathcal{P}^B}$ for the ambiguity set $\mathcal{P}^B$ satisfies:*

$$\mathbb{P}_{P^\star}\left[\hat{v}_{\mathcal{P}^B}^\pi \leq v_{P^\star}^\pi, \ \forall \pi \in \Pi \mid \mathcal{D}\right] \geq 1 - \delta .$$

*In addition, if $\hat{\pi}_{\mathcal{P}^B}^\star$ is the optimal solution to the RMDP, then $p_0^\mathsf{T} \hat{v}_{\mathcal{P}^B}^\star$ is a* safe *return estimate of $\hat{\pi}_{\mathcal{P}^B}^\star$.*

The proof is provided in Appendix B. Similar to other results, this theorem only proves that the constructed lower bound on the return is safe. It does not address the tightness of the bound.

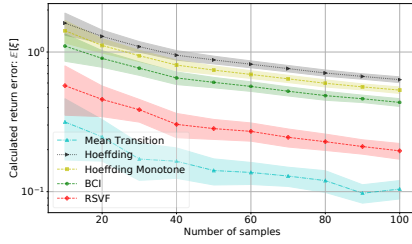
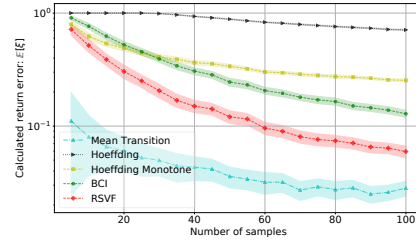

Figure 4: Expected regret of safe estimates with 95% confidence regions for the Bellman update with an uninformative prior.

Figure 5: Expected regret of safe estimates with 95% confidence regions for the Bellman update with an informative prior.

## 5 Empirical Evaluation

In this section, we empirically evaluate the safe estimates computed using Hoeffding, BCI, and RSVF ambiguity sets. We start by assuming a true model and generate simulated datasets from it. Each dataset is then used to construct an ambiguity set and a safe estimate of policy return. The performance of the methods is measured using the average of the absolute errors of the estimates compared with the true returns of the *optimal* policies. All of our experiments use a 95% confidence for the safety of the estimates.

We compare ambiguity sets constructed using BCI, RSVF, with the Hoeffding sets. To reduce the conservativeness of Hoeffding sets when transition probabilities are sparse, we use a modification inspired by the Good-Turing bounds [39]. That is that any transitions from $s, a$ to $s'$ are impossible if they are not in $\mathcal{D}$. We also compare with the "Hoeffding Monotone" formulation $\mathcal{P}^M$ even when there is no guarantee that the value function is really monotone. Finally, we compare the results with the "Mean Transition" which solves the expected model $\bar{p}_{s,a}$ with no safety guarantees.

Next in Section 5.1, we compare the methods in a simplified setting in which we consider the problem of estimating the value of a single state from a Bellman update. Then, Section 5.2, evaluates the approach on an MDP with an informative prior.

We do not evaluate the computational complexity of the methods since they target problems constrained by data and not computation. The Bayesian methods are generally more computationally demanding but the scale depends significantly on the type of the prior model used. All Bayesian methods draw $1,000$ samples from the posterior for each state and action.

### 5.1 Bellman Update

In this section, we consider a transition from a single state $s_0$ and action $a_0$ to 5 states $s_1, \ldots, s_5$. The value function for the states $s_1, \ldots, s_5$ is fixed to be $[1, 2, 3, 4, 5]$. RSVF is run for a single iteration with the given value function. The single iteration of RSVF in this simplistic setting helps to quantify the possible benefit of using RSVF-style methods over BCI. The ground truth is generated from the corresponding prior for each one of the problems.

**Uninformative Dirichlet Priors**  This setting considers a uniform Dirichlet distribution with $\alpha = [1, 1, 1, 1, 1]$ as the prior. This prior provides little information. Figure 4 compares the computed robust return errors. The value $\xi$ represents the regret of predicted returns, which is the absolute difference between the *true* optimal value and the robust estimate: $\xi = |\rho(\pi^\star_{P^\star}, P^\star) - \tilde{\rho}(\hat{\pi}^\star)|$. Here, $\tilde{\rho}$ is the robust estimate and $\hat{\pi}^\star$ is the optimal robust solution. The smaller the value, the tighter and less conservative the safe estimate is. The number of samples is the size of dataset $\mathcal{D}$. All results are computed by averaging over 200 simulated datasets of the given size generated from the true $P^\star$. The results show that BCI improves on both types of Hoeffding bounds and RSVF further improves on BCI. The mean estimate provides the tightest bounds, but it does not provide any meaningful safety guarantees.

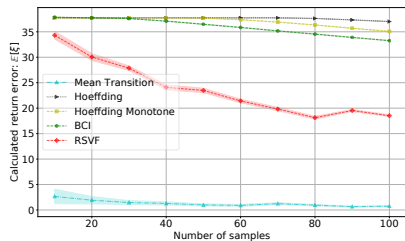
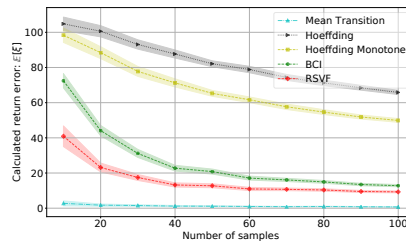

Figure 6: Expected regret of safe estimates with 95% confidence regions for the River-Swim: an MDP with an uninformative prior.

Figure 7: Expected regret of safe estimates with 90% confidence regions for the ExpPop-ulation: an MDP with an informative prior.

**Informative Gaussian Priors**    To evaluate the effect of using an informative prior, we use a problem inspired by inventory optimization. The states $s_1, \ldots, s_5$ represent inventory levels. The inventory level corresponds to the state index (1 in the state $s_1$) except that the inventory in the current state $s_0$ is 5. The demand is assumed to be Normally distributed with an unknown mean $\mu$ and a *known* standard deviation $\sigma = 1$. The prior over $\mu$ is Normal with the mean $\mu_0 = 3$ and, therefore, the posterior over $\mu$ is also Normal. The current action assumes that no product is ordered and, therefore, only the demand is subtracted from $s_0$.

## 5.2   Full MDP

In this section, we evaluate the methods using MDPs with relatively small state-spaces. They can be used with certain types of value function approximation, like aggregation [30], but we evaluate them only on tabular problems to prevent approximation errors from skewing the results. To prevent the sampling policy from influencing the results, each dataset $\mathcal{D}$ has the same number of samples from each state.

**Uninformative Prior**    We first use the standard RiverSwim domain for the evaluation [36]. The methods are evaluated identically to the Bellman update above. That is, we generate synthetic datasets from the ground truth and then compare the expected regret of the robust estimate with respect to the true return of the *optimal* policy for the ground truth. As the prior distribution, we use the uniform Dirichlet distribution over all states. Figure 6 shows the expected robust regret over 100 repetitions. The x-axis represents the number of samples in $\mathcal{D}$ for each state. It is apparent that BCI improves only slightly on the Hoeffding sets since the prior is not informative. RSVF, on the other hand, shows a significant improvement over BCI. All robust methods have safety violations of 0% indicating that even RSVF is unnecessarily conservative here.

**Informative Prior**    Next, we evaluate RSVF on the MDP model of a simple exponential population model [43]. Robustness plays an important role in ecological models because they are often complex, stochastic, and data collection is expensive. Yet, it is important that the decisions are robust due to their long term impacts. Figure 7 shows the average regret of safe predictions. BCI can leverage the prior information to compute tighter bounds, but RSVF further improves on BCI. The rate of safety violations is again 0% for all robust methods.

## 6   Summary and Conclusion

This paper proposes new Bayesian algorithms for constructing ambiguity sets in RMDPs, improving over standard distribution-free methods. BCI makes it possible to flexibly incorporate prior domains knowledge and is easy to generalize to other shapes of ambiguity sets (like $L_2$) without having to prove new concentration inequalities. Finally, RSVF improves on BCI by constructing tighter ambiguity sets that are not confidence regions. Our experimental results and theoretical analysis indicate that the new ambiguity sets provide much tighter safe return estimates. The only drawbacks of the Bayesian methods are that they need priors and may increase the computational complexity.

**Acknowledgments**

We would like to thank Vishal Gupta and the anonymous referees for their insightful comments and suggestions. This work was supported by NSF under grants number 1815275 and 1717368.

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
