[Supplementary Material · rsvf-app.pdf]

# A Technical Results

The following proposition shows that the guarantee of a safe estimate on the return is achieved when the true transition model is contained in the ambiguity set.

**Lemma A.1.** *Suppose that an ambiguity set $\mathcal{P}$ satisfies $\mathbb{P}_{\mathcal{D}}\left[p_{s,a}^{\star} \in \mathcal{P}_{s,a} \mid P^{\star}\right] \geq 1 - \delta/(SA)$ for each state $s$ and action $a$. Then:*

$$\mathbb{P}_{\mathcal{D}}\left[\hat{v}_{\mathcal{P}}^{\pi} \leq v_{P^{\star}}^{\pi}, \ \forall \pi \in \Pi \mid P^{\star}\right] \geq 1 - \delta .$$

*Proof.* We omit $\mathcal{P}$ and $P^{\star}$ from the notation in the proof since they are fixed. From Lemma B.1, we have that $\hat{v}^{\pi} \leq v^{\pi}$ if

$$\widehat{T}^{\pi}\hat{v}^{\pi} \leq T^{\pi}\hat{v}^{\pi} .$$

That is, for each state $s$ and action $a$:

$$\min_{p \in \mathcal{P}_{s,a}} p^{\mathsf{T}}\hat{v}^{\pi} \leq (p_{s,a}^{\star})^{\mathsf{T}}\hat{v}^{\pi}.$$

Using the identity above, the probability that the robust value function is a lower bound can be bounded as follows:

$$\mathbb{P}_{\mathcal{D}}\left[\hat{v}_{\mathcal{P}}^{\pi} \leq v_{P}^{\pi}, \ \forall \pi \in \Pi \mid P^{\star}\right] = \mathbb{P}_{\mathcal{D}}\left[\min_{p \in \mathcal{P}_{s,a}} p^{\mathsf{T}}\hat{v}^{\pi} \leq (p_{s,a}^{\star})^{\mathsf{T}}\hat{v}^{\pi}, \ \forall \pi \in \Pi, s \in \mathcal{S}, a \in \mathcal{A} \mid P^{\star}\right] \geq$$

$$\geq \mathbb{P}_{\mathcal{D}}\left[(p_{s,a}^{\star})^{\mathsf{T}}\hat{v}^{\pi} \leq (p_{s,a}^{\star})^{\mathsf{T}}\hat{v}^{\pi}, \ \forall \pi \in \Pi, s \in \mathcal{S}, a \in \mathcal{A} \mid P^{\star} \in \mathcal{P}, P^{\star}\right] \mathbb{P}_{\mathcal{D}}\left[P^{\star} \in \mathcal{P} \mid P^{\star}\right] +$$

$$+ \mathbb{P}_{\mathcal{D}}\left[P^{\star} \notin \mathcal{P} \mid P^{\star}\right] \geq 1\,\mathbb{P}_{\mathcal{D}}\left[P^{\star} \in \mathcal{P} \mid P^{\star}\right] + 0\,\mathbb{P}_{\mathcal{D}}\left[P^{\star} \notin \mathcal{P} \mid P^{\star}\right] \geq$$

$$\geq \mathbb{P}_{\mathcal{D}}\left[P^{\star} \in \mathcal{P} \mid P^{\star}\right] .$$

Now, from the union bound over all states and actions, we get:

$$\mathbb{P}_{\mathcal{D}}\left[\hat{v}^{\pi} > v^{\pi}|P^{\star}\right] \leq \mathbb{P}_{\mathcal{D}}\left[P^{\star} \notin \mathcal{P} \mid P^{\star}\right] \leq \sum_{s \in \mathcal{S}}\sum_{a \in \mathcal{A}} \mathbb{P}_{\mathcal{D}}\left[p_{s,a}^{\star} \notin \mathcal{P}_{s,a} \mid P^{\star}\right] \leq \delta ,$$

which completes the proof. $\qquad\square$

The next proposition is the Bayesian equivalent of Lemma A.1.

**Lemma A.2.** *Suppose that an ambiguity set $\mathcal{P}$ satisfies $\mathbb{P}_{P^{\star}}\left[p_{s,a}^{\star} \in \mathcal{P}_{s,a} \mid \mathcal{D}\right] \geq 1 - \delta/(SA)$ for each state $s$ and action $a$. Then:*

$$\mathbb{P}_{P^{\star}}\left[\hat{v}_{\mathcal{P}}^{\pi} \leq v_{P^{\star}}^{\pi}, \ \forall \pi \in \Pi \mid \mathcal{D}\right] \geq 1 - \delta .$$

*Proof.* We omit $\mathcal{P}$ and $P^{\star}$ from the notation in the proof since they are fixed. From Lemma B.1, we have that $\hat{v}^{\pi} \leq v^{\pi}$ if

$$\widehat{T}^{\pi}\hat{v}^{\pi} \leq T^{\pi}\hat{v}^{\pi} .$$

That is, for each state $s$ and action $a$:

$$\min_{p \in \mathcal{P}_{s,a}} p^{\mathsf{T}}\hat{v}^{\pi} \leq (p_{s,a}^{\star})^{\mathsf{T}}\hat{v}^{\pi}.$$

Using the identity above, the probability that the robust value function is a lower bound can be bounded as follows:

$$\mathbb{P}_{P^{\star}}\left[\hat{v}_{\mathcal{P}}^{\pi} \leq v_{P}^{\pi}, \ \forall \pi \in \Pi \mid \mathcal{D}\right] = \mathbb{P}_{P^{\star}}\left[\min_{p \in \mathcal{P}_{s,a}} p^{\mathsf{T}}\hat{v}^{\pi} \leq (p_{s,a}^{\star})^{\mathsf{T}}\hat{v}^{\pi}, \ \forall \pi \in \Pi, s \in \mathcal{S}, a \in \mathcal{A} \mid \mathcal{D}\right] \geq$$

$$\geq \mathbb{P}_{P^{\star}}\left[(p_{s,a}^{\star})^{\mathsf{T}}\hat{v}^{\pi} \leq (p_{s,a}^{\star})^{\mathsf{T}}\hat{v}^{\pi}, \ \forall \pi \in \Pi, s \in \mathcal{S}, a \in \mathcal{A} \mid P^{\star} \in \mathcal{P}, \mathcal{D}\right] \mathbb{P}_{P^{\star}}\left[P^{\star} \in \mathcal{P} \mid \mathcal{D}\right] +$$

$$+ \mathbb{P}_{P^{\star}}\left[P^{\star} \notin \mathcal{P} \mid \mathcal{D}\right] \geq 1\,\mathbb{P}_{P^{\star}}\left[P^{\star} \in \mathcal{P} \mid \mathcal{D}\right] + 0\,\mathbb{P}_{P^{\star}}\left[P^{\star} \notin \mathcal{P} \mid \mathcal{D}\right] \geq$$

$$\geq \mathbb{P}_{P^{\star}}\left[P^{\star} \in \mathcal{P} \mid \mathcal{D}\right] .$$

Now, from the union bound over all states and actions, we get:

$$\mathbb{P}_{P^{\star}}\left[\hat{v}^{\pi} > v^{\pi}|\mathcal{D}\right] \leq \mathbb{P}_{P^{\star}}\left[P^{\star} \notin \mathcal{P} \mid \mathcal{D}\right] \leq \sum_{s \in \mathcal{S}}\sum_{a \in \mathcal{A}} \mathbb{P}_{P^{\star}}\left[p_{s,a}^{\star} \notin \mathcal{P}_{s,a} \mid \mathcal{D}\right] \leq \delta ,$$

which completes the proof. $\qquad\square$

# B  Technical Proofs

## B.1  Proof of Theorem 3.1

Before presenting the proof of the theorem, we need to show some auxiliary results.

The following lemma shows that when the robust Bellman update lower-bounds the true Bellman update then the value function estimate is safe.

**Lemma B.1.** *Consider a policy $\pi$, its robust value function $\hat{v}^\pi$, and true value function $v^\pi$ such that $\hat{v}^\pi = \widehat{T}^\pi \hat{v}^\pi$ and $v^\pi = T^\pi v^\pi$. Then, $\hat{v}^\pi \leq v^\pi$ element-wise whenever $\widehat{T}^\pi \hat{v}^\pi \leq T^\pi \hat{v}^\pi$. That is, if $\min_{p \in \mathcal{P}_{s,a}} p^\mathsf{T} \hat{v}^\pi \leq p_{s,a}^\mathsf{T} \hat{v}^\pi$ for each state $s$ and action $a = \pi(s)$ then $\hat{v}^\pi \leq v^\pi$.*

Lemma B.1 implies readily that the inequality above is satisfied when $p_{s,a}^\star \in \mathcal{P}_{s,a}$.

*Proof.* Using the assumption $\widehat{T}^\pi \hat{v}^\pi \leq T^\pi \hat{v}^\pi$, and from $\hat{v}^\pi = \widehat{T}^\pi \hat{v}^\pi$ and $v^\pi = T^\pi v^\pi$, we get by algebraic manipulation:

$$\hat{v}^\pi - v^\pi = \widehat{T}^\pi \hat{v}^\pi - T_P^\pi v^\pi \leq T^\pi \hat{v}^\pi - T^\pi v^\pi = \gamma P_\pi (\hat{v}^\pi - v^\pi) \,.$$

Here, $P_\pi$ is the transition probability matrix for the policy $\pi$. Subtracting $\gamma P_\pi (\hat{v}^\pi - v^\pi)$ from the above inequality gives:

$$(\mathbf{I} - \gamma P_\pi)(\hat{v}^\pi - v^\pi) \leq \mathbf{0} \,,$$

where $\mathbf{I}$ is the identity matrix. Because the matrix $(\mathbf{I} - \gamma P_{\pi^\star})^{-1}$ is monotone, as can be seen from its Neumann series, we get:

$$\hat{v}^\pi - v^\pi \leq (\mathbf{I} - \gamma P_\pi)^{-1} \mathbf{0} = \mathbf{0} \,,$$

which proves the result. $\qquad\square$

The next lemma formalizes the safety-sufficiency of $\mathcal{K}$. Note that the rewards $r_{s,a}$ are not a factor in this lemma because they are certain and cancel out.

**Lemma B.2.** *Consider any ambiguity set $\mathcal{P}_{s,a}$ and a value function $v$. Then $\min_{p \in \mathcal{P}_{s,a}} p^\mathsf{T} v \leq (p_{s,a}^\star)^\mathsf{T} v$ with probability $1 - \delta/(SA)$ if and only if $\mathcal{P}_{s,a} \cap \mathcal{K}_{s,a}(v) \neq \emptyset$.*

*Proof.* To show the "if" direction, let $\hat{p} \in \mathcal{P}_{s,a} \cap \mathcal{K}_{s,a}(v)$. Such $\hat{p}$ exists because the intersection is nonempty. Then, $\min_{p \in \mathcal{P}_{s,a}} p^\mathsf{T} v \leq \hat{p}^\mathsf{T} v \leq \mathrm{V@R}_{P^\star}^\zeta \left[(p_{s,a}^\star)^\mathsf{T} v\right]$. By definition, $\mathrm{V@R}_{P^\star}^\zeta \left[(p_{s,a}^\star)^\mathsf{T} v\right] \leq (p_{s,a}^\star)^\mathsf{T} v$ with probability $1 - \delta/(SA)$.

To show the "only if" direction, suppose that $\hat{p}$ is a minimizer in $\min_{p \in \mathcal{P}_{s,a}} p^\mathsf{T} v$. The premise translates to $\mathbb{P}_{P^\star}[\hat{p}^\mathsf{T} v \leq (p_{s,a}^\star)^\mathsf{T} v \mid \mathcal{D}] \geq 1 - \delta/(SA)$. Therefore, $\mathrm{V@R}_{P^\star}^\zeta \left[(p_{s,a}^\star)^\mathsf{T} v\right] \geq \hat{p}^\mathsf{T} v$ and $\hat{p} \in \mathcal{P}_{s,a} \cap \mathcal{K}_{s,a}$ and the intersection is non-empty. $\qquad\square$

The following lemma formalizes the properties of $\mathcal{L}_{s,a}$.

**Lemma B.3.** *For any finite set $\mathcal{V}$ of value functions, the following inequality holds for all $v \in \mathcal{V}$ simultaneously:*

$$\mathbb{P}_{P^\star} \left[ \min_{p \in \mathcal{L}_{s,a}(\mathcal{V})} p^\mathsf{T} v \leq (p_{s,a}^\star)^\mathsf{T} v \;\middle|\; \mathcal{D} \right] \geq 1 - \frac{\delta}{SA} \,.$$

*Proof.* Assume an arbitrary $v \in \mathcal{V}$ and let $q_v^\star \in \arg\min_{q \in \mathcal{K}_{s,a}(v)} \|q - \theta_{s,a}(\mathcal{V})\|_1$ using the notation of (4). From the definition of $\theta_{s,a}(\mathcal{V})$ in (4), the value $q_v$ is in the ambiguity set $\mathcal{L}_{s,a}(\mathcal{V})$. Given that also $q_v \in \mathcal{K}_{s,a}(v)$, Lemma B.2 shows that:

$$\mathbb{P}_{P^\star} \left[ \min_{p \in \mathcal{L}_{s,a}(\mathcal{V})} p^\mathsf{T} v \leq (p_{s,a}^\star)^\mathsf{T} v \;\middle|\; \mathcal{D} \right] \geq 1 - \frac{\delta}{SA} \,,$$

because $q_v \in \mathcal{L}_{s,a}(v) \cup \mathcal{K}_{s,a}(v) \neq \emptyset$. This completes the proof since $v$ is any from $\mathcal{V}$. $\qquad\square$

We are now ready to prove the theorem.

*Proof.* Recall that Algorithm 1 terminates only if $\mathcal{K}_{s,a}(\hat{v}_k) \cap \mathcal{P}_{s,a}^k \neq \emptyset$ for each state $s$ and action $a$. Then, according to Lemma B.2, we get with probability $1 - \delta/(SA)$:

$$\min_{p \in \mathcal{P}_{s,a}^k} p^\mathsf{T} \hat{v}_k \leq (p_{s,a}^\star)^\mathsf{T} \hat{v}_k$$

for any fixed state $s$ and action $a$. By the union bound, the inequality holds simultaneously for all states and actions with probability $1 - \delta$. That means that with probability $1 - \delta$ we can derive the following using basic algebra:

$$\min_{p \in \mathcal{P}_{s,a}^k} p^\mathsf{T} \hat{v}_k \leq (p_{s,a}^\star)^\mathsf{T} \hat{v}_k \qquad \forall s \in \mathcal{S}, a \in \mathcal{A}$$

$$r_{s,a} + \min_{p \in \mathcal{P}_{s,a}^k} p^\mathsf{T} \hat{v}_k \leq r_{s,a} + (p_{s,a}^\star)^\mathsf{T} \hat{v}_k \qquad \forall s \in \mathcal{S}, a \in \mathcal{A}$$

$$\widehat{T}_{\mathcal{P}^k}^{\hat{\pi}_k} \hat{v}_k \leq T_{P^\star}^{\hat{\pi}_k} \hat{v}_k$$

Note that $\hat{v}_k$ is the robust value function for the policy $\hat{\pi}_k$ since $\hat{v}_k = \hat{v}_{\mathcal{P}_k}^\star$ and $\hat{\pi}_k = \hat{\pi}_{\mathcal{P}_k}^\star$. Lemma B.1 finally implies that $\hat{v}_k \leq v_{P^\star}^{\hat{\pi}_k}$ with probability $1 - \delta$. $\qquad \square$

## B.2 Proof of Theorem 4.1

*Proof.* The first part of the statement follows directly from Lemma A.1 and Lemma C.1. The second part of the statement follows from the fact that the lower bound property holds uniformly across all policies. $\qquad \square$

## B.3 Proof of Theorem 4.2

*Proof.* The first part of the statement follows directly from Lemma A.2 and the definition of $\psi_{s,a}^B$. The second part of the statement follows from the fact that the lower bound property holds uniformly across all policies. $\qquad \square$

# C $L_1$ Concentration Inequality Bounds

In this section, we describe a new elementary proof of a bound on the $L_1$ distance between the estimated transition probability distribution and the true one. It simplifies the proofs of Weissman et al. (2003) but also leads to coarser bounds. We include the proof here in order to derive the tighter bound in Appendix C.1. Note that in the frequentist setting the ambiguity set $\mathcal{P}$ is a random variable that is a function of the dataset $\mathcal{D}$.

Recall that our ambiguity sets are defined as $L_1$ balls around the expected transition probabilities $\bar{p}_{s,a}$:

$$\mathcal{P}_{s,a} = \{p \in \Delta^S : \|p - \bar{p}_{s,a}\|_1 \leq \psi_{s,a}\} . \tag{6}$$

Lemma A.1 implies that the size of the $L_1$ balls must be chosen as follows:

$$\mathbb{P}\left[\|\bar{p}(s,a) - p^\star(s,a)\|_1 \leq \psi_{s,a}\right] \geq 1 - \delta/(SA) . \tag{7}$$

We can now express the necessary size $\psi_{s,a}$ of the ambiguity sets in terms of $n_{s,a}$, which denotes the number of samples in $\mathcal{D}$ that originate with a state $s$ and an action $a$.

**Lemma C.1** ($L_1$ Error bound). *Suppose that $\bar{p}_{s,a}$ is the empirical estimate of the transition probability obtained from $n_{s,a}$ samples for each $s \in \mathcal{S}$ and $a \in \mathcal{A}$. Then:*

$$\mathbb{P}\left[\|\bar{p}_{s,a} - p_{s,a}^\star\|_1 \geq \psi_{s,a}\right] \leq (2^S - 2)\exp\left(-\frac{\psi_{s,a}^2 n_{s,a}}{2}\right) .$$

*Therefore, for any $\delta \in [0,1]$:*

$$\mathbb{P}\left[\|\bar{p}_{s,a} - p_{s,a}^\star\|_1 \leq \sqrt{\frac{2}{n_{s,a}} \log \frac{SA(2^S - 2)}{\delta}}\right] \leq 1 - \delta/(SA) .$$

*Proof.* To shorten the notation, we omit the indexes $s, a$ throughout the proof; for example $\bar{p}$ is used instead of the full $\bar{p}_{s,a}$. First, express the $L_1$ distance between two distributions $\bar{p}$ and $p^\star$ in terms of an optimization problem. Let $\mathbf{1}_{\mathcal{Q}} \in \mathbb{R}^S$ be the indicator vector for some subset $\mathcal{Q} \subset \mathcal{S}$. Then:

$$\|\bar{p} - p^\star\|_1 = \max_z \left\{ z^\mathsf{T}(\bar{p} - p^\star) \ : \ \|z\|_\infty \leq 1 \right\} =$$
$$= \max_{\mathcal{Q} \in 2^S} \left\{ \mathbf{1}_{\mathcal{Q}}^\mathsf{T}(\bar{p} - p^\star) - (\mathbf{1} - \mathbf{1}_{\mathcal{Q}})^\mathsf{T}(\bar{p} - p^\star) \ : \ 0 < |\mathcal{Q}| < m \right\}$$
$$\overset{\text{(a)}}{=} 2 \max_{\mathcal{Q} \in 2^S} \left\{ \mathbf{1}_{\mathcal{Q}}^\mathsf{T}(\bar{p} - p^\star) \ : \ 0 < |\mathcal{Q}| < m \right\} \ .$$

Here, (a) holds because $\mathbf{1}^\mathsf{T}(\bar{p} - p^\star) = 0$. Using the expression above, the target probability can be bounded as follows:

$$\mathbb{P}\left[\|\bar{p} - p^\star\|_1 > \psi\right] = \mathbb{P}\left[ 2 \max_{\mathcal{Q} \in 2^S} \left\{ \mathbf{1}_{\mathcal{Q}}^\mathsf{T}(\bar{p} - p^\star) \ : \ 0 < |\mathcal{Q}| < m \right\} > \psi \right]$$
$$\overset{\text{(a)}}{\leq} (|\mathcal{Q}| - 2) \max_{\mathcal{Q} \in 2^S} \left\{ \mathbb{P}\left[ \mathbf{1}_{\mathcal{Q}}^\mathsf{T}(\bar{p} - p^\star) > \frac{\psi}{2} \right] \ : \ 0 < |\mathcal{Q}| < m \right\}$$
$$\overset{\text{(b)}}{\leq} (|\mathcal{Q}| - 2) \exp\left( -\frac{\psi^2 n}{2} \right) = (2^S - 2) \exp\left( -\frac{\psi^2 n}{2} \right) \ .$$

The inequality (a) follows from union bound and the inequality (b) follows from the Hoeffding's inequality since $\mathbf{1}_{\mathcal{Q}}^\mathsf{T}\bar{p} \in [0, 1]$ for any $\mathcal{Q}$ with the mean of $\mathbf{1}_{\mathcal{Q}}^\mathsf{T}\bar{p}^\star$. $\qquad \square$

### C.1 Ambiguity Sets for Monotone Value Functions

A significant limitation of the result in Lemma C.1 is that the $\psi$ depends linearly on the number of states. We now explore an assumption that can alleviate this important drawback when the value functions are guaranteed to be monotone. In particular, the monotonicity assumption states that the value functions $v$ of the optimal robust policy must be non-decreasing in some arbitrary order which must be known ahead of time. Assume, therefore, without loss of generality that:

$$v_1 \geq v_2 \geq \ldots \geq v_n \ , \tag{8}$$

where $v_i$ is the value of state $i$.

Admittedly, monotonicity is a restrictive assumption, but we explore it in order to understand the greatest possible gains from tightening the known concentration inequalities. Yet, monotonicity of this type occurs in some problems, such as inventory management in which the value does not decrease with increasing inventory levels or medical problems in which the value does not increase with a deteriorating health state.

It is important to note that any MDP algorithm that relies on the assumption (8) needs to also enforce it. That means, the algorithm must prevent generating value functions that violate the monotonicity assumption. Practically, this could be achieved by representing the value function as a linear combination of monotone features.

The bound Lemma C.1 is large because of the term $2^S$ which derives from the use of a union bound. The union bound is used because the $L_1$ norm can be represented as a maximum over an exponentially many linear functions:

$$\|x\|_1 = \max_{\mathcal{Q} \subseteq \mathcal{I}} \left( \mathbf{1}_{\mathcal{Q}} - \mathbf{1}_{\mathcal{I} \setminus \mathcal{Q}} \right)^\mathsf{T} x \ .$$

Here, the set $\mathcal{I} = 2^S$ represents all indexes of $x$ and $\mathbf{1}_{\mathcal{Q}}$ is a vector that is one for all elements of $\mathcal{Q}$ and zero otherwise. We now show that under the monotonicity property (8), the $L_1$ norm can be represented as a maximum over a *linear* (in states) number of linear functions. In particular, the worst-case optimization problem of the nature:

$$
\begin{aligned}
\min_p \quad & v^\mathsf{T} p \\
\text{s.t.} \quad & \left( \mathbf{1}_{\mathcal{Q}} - \mathbf{1}_{\mathcal{I} \setminus \mathcal{Q}} \right)^\mathsf{T} (p - \bar{p}) \leq \psi, \quad \forall \mathcal{Q} \subseteq \mathcal{I} \\
& \mathbf{1}^\mathsf{T} p = 1, \\
& p \geq 0
\end{aligned}
\tag{9}
$$

can be replaced by the following optimization problem:

$$
\begin{aligned}
\min_{p} \quad & v^{\mathsf{T}} p \\
\text{s.t.} \quad & (\mathbf{1}_{k\dots n} - \mathbf{1}_{1\dots(k-1)})^{\mathsf{T}}(p - \bar{p}) \leq \psi, \quad \forall k = 0, \dots, (n+1) \\
& \mathbf{1}^{\mathsf{T}} p = 1, \\
& p \geq 0
\end{aligned}
\tag{10}
$$

**Lemma C.2.** *Suppose that* (8) *is satisfied. Then the optimal objective values of* (9) *and* (10) *coincide.*

*Proof.* Let $f^a$ be the optimal objective of (9) and let $f^b$ be the optimal objective of (10). The inequality $f^a \geq f^b$ can be shown readily since (10) only relaxes some of the constraints of (9).

It remains to show that $f^a \leq f^b$. To show the inequality by contradiction, assume that each optimal solution $p^b$ to (10) is infeasible in (9) (otherwise $f^a \leq f^b$). Let the constraint violated by $p^b$ be:

$$
\left(\mathbf{1}_{\mathcal{C}} - \mathbf{1}_{2^S \setminus \mathcal{C}}\right)^{\mathsf{T}}(p - \bar{p}) \leq \psi,
$$

for some set $\mathcal{C}$. Since this constraint is not present in (10), that means that there exist $i$ and $j$ such that $i < j$, $i \in \mathcal{C}$, $j \notin \mathcal{C}$, and because the constraint is violated:

$$
p_i^b = \bar{p}_i - \epsilon, \quad \text{or} \quad p_j^b = \bar{p}_j + \epsilon
$$

for some $\epsilon > 0$. Assume now that $p_i^b = \bar{p}_i - \epsilon$, the case when $p_j^b = \bar{p}_j + \epsilon$ follows similarly.

Now, choose the largest $k > i$ possible, and let $p^a = p^b$, with the exception of:

$$
p_i^a = p_i^b + \epsilon, \quad \text{and} \quad p_k^a = p_k^b - \epsilon.
$$

This does not increase the violation of the constraint by $p^a$ over $p^b$:

$$
\left(\mathbf{1}_{\mathcal{C}} - \mathbf{1}_{2^S \setminus \mathcal{C}}\right)^{\mathsf{T}}(p^a - \bar{p}) \leq \left(\mathbf{1}_{\mathcal{C}} - \mathbf{1}_{2^S \setminus \mathcal{C}}\right)^{\mathsf{T}}(p^b - \bar{p}),
$$

And it does not increase the objective function:

$$
v^{\mathsf{T}} p^a = v^{\mathsf{T}} p^b - \epsilon(v_i - v_j) \leq v^{\mathsf{T}} p^b,
$$

and thus remains optimal in (10). Repeating these steps until no constraints are violated leads to a contradiction with the lack of an optimal solution to (10) that is not optimal in (9). □

Lemma C.2 shows that we can replace the $L_1$ ambiguity set in (6) by the following set without affecting the solution.

$$
\mathcal{P}_{s,a} = \{p \in \Delta^S \ : \ (\mathbf{1}_{k\dots n} - \mathbf{1}_{1\dots(k-1)})^{\mathsf{T}}(p - \bar{p}_{s,a}) \leq \psi_{s,a}, \quad \forall k = 0, \dots, (n+1)\}
\tag{11}
$$

Now, following the same steps as the proof of Lemma C.1 but using (11) in place of (6) gives us the following result.

**Lemma C.3** ($L_1$ Error bound). *Suppose that $\bar{p}_{s,a}$ is the empirical estimate of the transition probability obtained from $n_{s,a}$ samples for each $s \in \mathcal{S}$ and $a \in \mathcal{A}$. Then:*

$$
\mathbb{P}\left[\|\bar{p}_{s,a} - p_{s,a}^\star\|_1 \geq \psi_{s,a}\right] \leq S \exp\left(-\frac{\psi_{s,a}^2 n_{s,a}}{2}\right).
$$

*Therefore, for any $\delta \in [0, 1]$:*

$$
\mathbb{P}\left[\|\bar{p}_{s,a} - p_{s,a}^\star\|_1 \leq \sqrt{\frac{2}{n_{s,a}} \log \frac{S^2 A}{\delta}}\right] \leq 1 - \delta/(SA).
$$

# D    Detailed Description of Selected Algorithms

## D.1    Computing Bayesian Confidence Interval

---

**Algorithm 2:** Bayesian Confidence Interval (BCI)

---

    **Input:** Distribution $\theta$ over $p^\star_{s,a}$, confidence level $\delta$, sample count $m$
    **Output:** Nominal point $\bar{p}_{s,a}$ and $L_1$ norm size $\psi_{s,a}$

**1**  Sample $X_1, \ldots, X_m \in \Delta^S$ from $\theta$: $X_i \sim \theta$;
**2**  Nominal point: $\bar{p}_{s,a} \leftarrow (1/m) \sum_{i=1}^{m} X_i$;
**3**  Compute distances $d_i \leftarrow \|\bar{p}_{s,a} - X_i\|_1$ and sort *increasingly*;
**4**  Norm size: $\psi_{s,a} \leftarrow d_{(1-\delta)\,m}$;
**5**  **return** $\bar{p}_{s,a}$ *and* $\psi_{s,a}$;

---

# E    Why Not Credible Regions

Constructing ambiguity sets from confidence regions seems intuitive and natural. It may be surprising that RSVF abandons this intuitive approach. In this section, we describe two reasons why confidence regions are unnecessarily conservative compared to RSVF sets.

The first reason why confidence regions are too conservative is because they assume that the value function depends on the true model $P^\star$. To see this, consider the setting of Example 2.1 with $r_{s_1,a_1} = 0$. When an ambiguity set $\mathcal{P}_{s_1,a_1}$ is built as a confidence region such that $\mathbb{P}[p^\star_{s_1,a_1} \in \mathcal{P}_{s_1,a_1}] \geq 1 - \delta$, it satisfies:

$$\mathbb{P}_{P^\star} \left[ \min_{p \in \mathcal{P}_{s,a}} p^\mathsf{T} v \leq (p^\star_{s,a})^\mathsf{T} v, \ \forall v \in \mathbb{R}^S \ \middle| \ \mathcal{D} \right] \geq 1 - \delta.$$

Notice the value function inside of the probability operator. Lemma B.1 shows that this guarantee is needlessly strong. It is, instead, sufficient that the inequality in Lemma B.1 holds just for $\hat{v}^\pi$ which is independent of $P^\star$ in the Bayesian setting. The following weaker condition is sufficient to guarantee safety:

$$\mathbb{P}_{P^\star} \left[ \min_{p \in \mathcal{P}_{s,a}} p^\mathsf{T} v \leq (p^\star_{s,a})^\mathsf{T} v \ \middle| \ \mathcal{D} \right] \geq 1 - \delta, \ \forall v \in \mathbb{R}^S \tag{12}$$

Notice that $v$ is outside of the probability operator. This set is smaller and provides the same guarantees, but may be more difficult to construct [13].

The second reason why confidence regions are too conservative is because they construct a uniform lower bound for all policies $\pi$ as is apparent in Theorem 4.2. This is unnecessary, again, as Lemma B.1 shows. The robust Bellman update only needs to lower bound the Bellman update for the computed value function $\hat{v}^\pi$, not for all value functions. As a result, (12), can be further relaxed to:

$$\mathbb{P}_{P^\star} \left[ \min_{p \in \mathcal{P}_{s,a}} p^\mathsf{T} \hat{v}^{\pi_R} \leq (p^\star_{s,a})^\mathsf{T} \hat{v}^{\pi_R} \ \middle| \ \mathcal{D} \right] \geq 1 - \delta, \tag{13}$$

where $\pi_R$ is the optimal solution to the robust MDP. RSVF is less conservative because it constructs ambiguity sets that satisfy the weaker requirement of (13) rather than confidence regions. Deeper theoretical analysis of the benefits of using RSVF sets is very important but is beyond the scope of this work. Examples that show the benefits to be arbitrarily large or small can be constructed readily by properly choosing the priors over probability distributions.