[Reviews · NeurIPS 2019]

Reviewer 1



*** Post-rebuttal *** Thanks for the authors' response and other reviews. In general all the reviewers seem to agree that the paper often lacks intuition for the proposed algorithm but the contributions look quite significant. It was also great for the authors to notice the analysis on computational complexity was unintentionally missing. Though I was leaning towards acceptance, I'd like to retain my original score given the confidence about my assessment was not super strong. This paper proposes a novel Bayesian approach, RSVF (Robustification with Sensible Value Functions), to building an ambiguity set for a RMDP. A major limitation on applicability of RMDP approaches has been the looseness of ambiguity set given a robustness constraint resulting an overly conservative agent behaviour. Hence the study on tighter ambiguity set is a welcome contribution to the literature. The intuition behind RSVF, as described in Section E in appendix, is based on two observations on the looseness of existing approaches to confidence regions. Roughly speaking, the loose bound is caused by unnecessary strong conditions enforced to all possible value functions and policies. These findings inspire the proposed algorithm, RSVF, which iteratively update the set of candidate optimal value functions and derives the approximately optimal ambiguity set for those value functions. The algorithm unfortunately does not guarantee to converge, but it guarantees the return policy and its corresponding return estimate to be safe. In the experiment section RSVF has been compared to a several of approaches to building ambiguity sets, such as one based on Hoeffding bound and Bayesian credible/confidence interval. In both cases of single Bellman update and full MDP, RSVF turns out to build tighter ambiguity sets than others though it still displayed unnecessary conservatism (zero safety violation) and hence there is a room for being further tightened. There might be some criticisms though. As pointed out in the paper, the computational complexity is not studied or evaluated so the practicality of this approach might look questionable. The paper is clearly written in general, but it is very hard to understand the algorithm without reading the appendix or references. Eq (4) could benefit from illustrative examples, in addition to Figure 1-3.

Reviewer 2



This work describes a novel approach for constructing ambiguity sets for robust MDPs, based on Bayesian techniques. Robust MDPs are an interesting and important area of research which allow us to learn policies with high-confidence guarantees of success. This new algorithm builds on past work, providing an algorithm which allows us to compute tighter bounds on the final performance of the policy. The paper is overall fairly well written and structured, and thoroughly references relevant background material. Several minor typos scattered throughout. One place that the clarity of this work is lacking is in intuition for the proposed algorithm. For example, sets K and L are formally laid out, but the prose does not inform the reader of why they must be defined this way. The figures (1, 2, and 3) are a good attempt, but they are poorly described, and thus do not do a good job conveying intuitions. Figures more directly tied to the proposed algorithm would be nice as well; for example, a visualization of how the POV might evolve throughout training. The authors do a good job justifying their approach on an intuitive level, particularly in Appendix E. (Which I would have liked to see this highlighted in the main paper.) However, the theoretical and empirical evidence for their approach is somewhat lacking. There is no principled analysis of the tightness of this method relative to other approaches, and experiments consist of only some very basic grid-world experiments against simple baselines. The approach performs uniformly worse than a generic non-safe baseline on all tasks, which means that the experiments are inadequate for showing effectiveness. The authors should choose some experimental settings for which safety is important (for example, by dramatically increasing the state space relative to the data, or by having a bad prior), and demonstrate that not only do safe methods outperform MLE methods, but *also* RSVF outperforms baselines. One high-level criticism I have with this work is that any Bayesian approach to safety is tightly bound to the accuracy of the prior. Safety guarantees are primarily useful when they are applicable in any situation, to any MDP, regardless of the user's knowledge of the problem. This is the case for concentration-inequality based bounds, which are very general. If I am not mistaken, the bounds in this paper rely on P* being a true posterior for the data, which in turn relies on the prior being accurate. But if we have an accurate prior, and the ability to do efficient posterior updates (or at least efficient posterior sampling), then we can just solve for the Bayes-optimal policy on the distribution of MDPs sampled from the posterior, and get optimality guarantees for that: no need to bother with robustness. Is this interpretation correct? If so, what do robustness methods add in this setting? ---------- EDIT: Thanks to the authors for their response. I feel that the authors have a good understanding of my concerns, and that they will be addressed in the final copy. I recommend acceptance.

Reviewer 3



Originality: To the best of my knowledge, constructing ambiguity sets beyond confidence bounds and credible regions is new to this work. However, there might be sampling based heuristic techniques that closely resemble such methods. Significance: It is hard to assess the significance of this work without a generalization bound or further empirical evaluation. As the paper mentions, the resulting solution is only guaranteed to be safe, and not necessarily accurate. The computational complexity is also not established, pushing the analysis further into speculations. Quality and correctness: The analysis presented in the paper look correct to me. I did not fully check the proofs in the appendix. Clarity: The paper is dense in parts and the language is sometimes confusing. The notation is not easy to follow (a notation table could help).

[Author Response · NeurIPS 2019]

We would like to thank the reviewers for their time and helpful comments. We will clarify/fix the paper as suggested.

**Reviewer #1**

> *the computational complexity is not studied or evaluated so the practicality of this approach might look questionable.*

Thank you for pointing that out. Note that all algorithms we proposed run in quasi-linear time in the number states, actions, samples, and $1/\gamma$ (particularly solving the RMDPs). We unintentionally omitted this discussion from the paper and we will rectify the omission. Also, the Batch-RL setup is constrained by samples and not computational complexity.

> *It would be great to improve the readability by having intuitive explanations and illustrative examples.*

We agree. There was a tradeoff in writing and explaining the ideas while satisfying the page limit constraints. We will add a very simple clarifying example as suggested.

**Reviewer #2**

> *sets $\mathcal{K}$ and $\mathcal{L}$ are formally laid out, but the prose does not inform the reader of why they must be defined this way.*

Thank you for pointing out this omission. We will add the intuition for these sets. Briefly, the set $\mathcal{K}$ is the set of probability distributions which, if contained in the ambiguity set, are sufficient to guarantee the safety of the solution.

> *Figures more directly tied to the proposed algorithm would be nice as well; for example, a visualization of how the POV might evolve throughout the training.*

Figure 3 is an attempt to demonstrate just that. We agree that a better visualization could help to better explain it, but we were not able to add one due to the page constraints. We will add such an illustration in the supplementary material.

> *experiments consist of only some very basic grid-world experiments against simple baselines*

We agree, we also think that future work should obtain results on bigger domains. The paper focuses on simple domains to avoid compounding errors from value function approximation and modeling.

> *The approach performs uniformly worse than a generic non-safe baseline on all tasks, which means that the experiments are inadequate for showing effectiveness.*

The non-safe baseline is not comparable to the other methods since it does not provide any guarantees. We concur that including the non-safe method in the same plot is confusing and will remove it.

> *But if we have an accurate prior, and the ability to do efficient posterior updates (or at least efficient posterior sampling), then we can just solve for the Bayes-optimal policy on the distribution of MDPs sampled from the posterior, and get optimality guarantees for that: no need to bother with robustness. Is this interpretation correct? If so, what do robustness methods add in this setting?*

That is an interesting point. The problem is that the Bayes-optimal solution requires solving a POMDP (difficult) and does not offer the required performance guarantees. The robust solution is easier to compute and provides the requisite performance guarantees. Our work can also be seen as a risk-averse solution for the Bayes-MDP but that is a topic for a different paper.

**Reviewer #3**

> *The computational complexity is also not established, pushing the assessment further into speculations.*

We agree that the analysis of computational complexity is important. Note that all algorithms we proposed run in quasi-linear time in the number states, actions, samples, and $1/\gamma$ (particularly solving the RMDPs). The only unknown is the number of iterations necessary, but in all our experiments the optimization converges in less than 10 iterations. We will emphasize this point in the paper.

> *The paper is dense in parts and the language is sometimes confusing. The notation is not easy to follow.*

We will make the language more precise and add a notation table as suggested.

[Meta-Review · NeurIPS 2019]

The paper proposes a novel approach for robust MDPs. The reviewers found the paper interesting, but difficult to read. The theoretical contribution is significant and the empirical contribution is adequate to support the theory. If the paper is accepted, the authors are strongly encouraged to follow the advice of the reviewers to improve the readability of the paper.